# Weirdo19ES is a novel singleton mycobacteriophage that selects for glycolipid deficient phage-resistant *M. smegmatis* mutants

**Cristian Alejandro Suarez, Jorgelina Judith Franceschelli, Sabrina Emilse Tasselli, Héctor Ricardo Morbidoni**◉*

Laboratorio de Microbiología Molecular, Facultad de Ciencias Médicas, Universidad Nacional de Rosario, Rosario, Argentina

* morbiatny@yahoo.com

**Data Availability Statement:** We declare that the data that support the findings reported in this manuscript are available in NCBI at https://www.ncbi.nlm.nih.gov/, Accesion number MN103533.

## Abstract

The sequencing and bioinformatics analysis of bacteriophages infecting mycobacteria has yielded a large amount of information on their evolution, including that on their environmental propagation on other genera such as *Gordonia*, closely related to *Mycobacterium*. However, little is known on mycobacteriophages cell biology such as the nature of their receptor (s) or their replication cycle. As part of our on-going screening for novel mycobacteriophages, we herein report the isolation and genome bioinformatics analysis of Weirdo19ES, a singleton Siphoviridae temperate mycobacteriophage with a 70.19% GC content. Nucleotide and protein sequence comparison to actinobacteriophage databases revealed that Weirdo19ES shows low homology to *Gordonia* phage Ruthy and mycobacteriophages falling in clusters Q and G and to singleton DS6A.Weirdo19ES also displays uncommon features such as a very short Lysin A gene (with only one enzymatic domain) and two putative HNH endonucleases. *Mycobacterium smegmatis* mutants resistant to Weirdo19ES are cross- resistant to I3. In agreement with that phenotype, analysis of cell envelope of those mutants showed that Weirdo19ES shares receptors with the transducing mycobacteriophage I3.This singleton mycobacteriophage adds up to the uncommonness of local mycobacteriophages previously isolated by our group and helps understanding the nature of mycobacteriophage receptors.

## Introduction

Bacteriophages infecting the genus *Mycobacterium* are to this day, the best characterized bacterial viruses with more than 1,700 genomes sequenced and roughly 12,000 phages isolated. Several reports have described the development of the bioinformatics programs needed for mycobacteriophage analysis, as well as the construction of mycobacteriophage based tools made for genetic manipulation of mycobacteria such as the *recombineering* method to modify mycobacterial and mycobacteriophage genomes [1, 2]. Taken collectively, the information

These data were derived from the following resources available in the public domain: The Actinobacteriophage database at https://phagesdb.org/phages/Webster/.

**Funding:** HRM has received grant ANPCyT PICT 3795 (MinCyT, Argentina) https://www.argentina.gob.ar/ciencia/agencia/fondo-para-la-investigacion-cientifica-y-tecnologica-foncyt The funder played no role in the study, the preparation of the article or the decision to publish.

**Competing interests:** I have read the journal's policy and the authors of this manuscript have no competing interests.

produced through genome sequence analysis gives an ample view of the scenario of mycobacteriophage evolution. As an example, the characterization of mycobacteriophage Patience revealed a low GC content (53%) and codon usage that did not reflect those of its laboratory host, *Mycobacterium smegmatis* but was more related to *Mycobacterium abscessus* [3]. Similarly, phages infecting *Gordonia* species showed homology to mycobacteriophages suggesting propagation across genera [4].

Several mycobacteriophages were isolated and characterized by our group with the aim of exploring their diversity at a geographical location (South America) in which little work on that matter had been carried out [5–7]. Mycobacteriophages defining novel clusters and subclusters were detected by our group in spite of the reduced number of total isolates (less than 30) [8]. One of such phages, Weirdo19ES (initially named 19ES) was isolated in Argentina from soil samples during a previous screening. Preliminary characterization by TEM of Weirdo19ES showed that it has a Siphoviridae morphology with a length of approximately 250 nm and an icosahedral capsid of 70 nm. This phage produced small turbid plaques at 30°C and larger plaques at 37°C but failed to propagate at 42°C, behaved as temperate and failed to infect *M. tuberculosis* [5]. Surprisingly Weirdo19ES propagated in *M. smegmatis* without addition of extra calcium or magnesium cations to the growth medium, a common requirement for most mycobacteriophages [5]. Preliminary analysis of Weirdo19ES genome sequence indicated that it was a singleton (Franceschelli, J.J., personal communication). In sum, those features of this mycobacteriophage led us to pursue its full bioinformatics analysis, the characterization of its life cycle and nature of its mycobacterial receptor(s), results that are reported herein.

## Materials and methods

### Strains, chemicals, culture media and growth conditions

*Mycobacterium smegmatis* mc$^2$155 (lab stock) and derivatives were grown at 37°C in Middlebrook 7H9 broth supplemented with 0.5% (w/v) glycerol, 10% ADS (albumin-dextrose-NaCl) and 0.2% (w/v) Tween 80 (hereafter designated as 7H9 ADS Gly Tw for short) as liquid culture medium. The same medium devoid of Tween 80 and supplemented with agar 1.5% (w/v) was as solid medium (7H9 ADS Gly agar). Mutant strain *M. smegmatis* MSMEG_0398::Tn61 (Myc55), -a glycopeptidolipid (GPL) deficient mutant generated through insertional transposon mutagenesis- was the generous gift of Dr. M. Daffé (Institut de Pharmacologie et de Biologie Structurale, Toulouse). Mycobacteriophages I3 and D29, used as controls throughout this study, were lab stocks. All chemicals and solvents were purchased from Sigma (St Louis, Mo) unless otherwise stated.

### General bacteriophage techniques

Preparation of mycobacteriophage high titer lysates and their titer determination were done as previously published [5]; briefly, late-log phase culture of *M. smegmatis* mc$^2$155 grown at 37°C were spun down and washed twice in Phage Buffer (PhB; 50 mM Tris-HCl pH 7.6, 150 mM NaCl, 2 mM CaCl$_2$, 10 mM MgSO$_4$) and resuspended in the same buffer to the original volume. Aliquots containing the desired number of Plaque Forming Units (PFU) (usually 10$^5$ PFU) of the phages under assay were mixed with 100 μl of the washed *M. smegmatis* cells for 30 min at room temperature, followed by addition of 4 ml of molten warm top agar (0.4% agar in 7H9 broth containing 0.5% glycerol (w/v) and 2 mM CaCl$_2$), gentle mixing and finally poured on top of Middlebrook 7H9 0.5% (w/v) glycerol, 2 mM CaCl$_2$ agar plates (hereafter designated as indicator plates, IP). After 48–72 h incubation at 37°C, phages on those plates showing confluent lysis were eluted with PhB at 4°C for 12–24 h, the liquid collected, centrifuged and the clean supernatant filter sterilized. For phage titer determination, molten warm

top agar (4 ml) were mixed with 100 μl of *M. smegmatis* washed cells and poured onto IP plates. After hardening of the top agar, 10 μl of 1/10 dilutions of each lysate in PhB were spotted on the top agar surface, plates were dried at room temperature and incubated at 37˚C for 48–72 h after which titers were calculated by visual inspection. Lysates were kept at 4˚C.

The determination of the latency period was carried out as described by McNerney et al [9]; briefly, 100 μl of stationary cultures of *M. smegmatis* mc$^2$155 were diluted to $10^8$ CFU /ml and infected with phage at a m.o.i ≈ 0.01. The mixtures were maintained in a water bath at 37˚C and at regular time intervals following mixing (time zero), 10 μl aliquots were taken for plating on IP; after overnight incubation at 37˚C the numbers of plaques produced were counted. The latent period was determined as the time at which an increase in the numbers of PFU was seen on the IP.

Lysogenization frequencies were determined by plating *M. smegmatis* mc$^2$155 cells ($10^9$ CFU) onto plates seeded with aliquots containing $10^9$–$10^{10}$ PFU of Weirdo19ES. Clones surviving on those plates were isolated and purified; their lysogenic nature was assayed by UV induction as previously described [5]. Frequency of lysogenization was calculated as the number of clones releasing phage/number total cells in the assay.

## Isolation and characterization of mycobacteriophage resistant *M. smegmatis* mutants

In order to obtain spontaneous mutants resistant to mycobacteriophages, 1 ml aliquots of late-log cultures of *M. smegmatis* were washed twice in PhB and resuspended in the same buffer to the original volume. Small aliquots (100 μl) were plated on top of 7H9 ADS Gly agar plates which had previously been seeded with $10^9$–$10^{10}$ PFU/ml of mycobacteriophages I3 or Weirdo19ES; afterwards plates were incubated at 37˚C for 5 days. Ten colonies arising on each plate, representing possible survivors to each mycobacteriophage, were picked and purified twice by streaking on fresh 7H9 ADS Gly agar plates. Resistance to the mycobacteriophage used as selection agent was tested by growing each colony to late-log phase in 5 ml 7H9 ADS Gly Tw, washing the cultures twice with PhB and resuspending them in the same buffer to the original volume. Each clone was challenged against the mycobacteriophage used for its selection by a cross-streaking method, followed by incubation at 37˚C for 3 days. Clones that grew without disturbance after contacting the phage were purified on 7H9 Gly plates; to eliminate possible lysogens, each clone was grown again in 7H9 Gly liquid medium and 100 μl aliquots were tested by UV induction as described [5]. Those clones in which UV light induced lysis were considered lysogens and kept for future studies (not included in this report). Finally, the putative spontaneous resistant mutants were used as indicator strains against the mycobacteriophages used for their selection to confirm the resistance phenotype and also tested for cross-resistance to the other mycobacteriophages used in this study.

In order to detect differences in colony morphology, each mycobacteriophage resistant clone was plated on 7H9 ADS Gly plates containing Congo Red (100 μg ml-1) and incubated for 3–4 days at 37˚C [10]. Colony morphology was assessed by visual inspection and pictures taken at a magnification of 8X with an Olympus MVX10 microscope; differences in morphotypes were recorded.

**Aggregation assay.** In order to determine the extent of cell-cell interaction in wild-type and mycobacteriophage resistant *M. smegmatis* mutants, each strain was grown to early stationary phase in 7H9 ADS Gly, followed by spontaneous sedimentation (1xg) for 10 min to separate aggregates as described [11]. The supernatants were gently transferred to new tubes and their $OD_{600}$ was measured. The cell aggregates in the sediments were broken up by briefly shaking them by vortex with 4 mm glass beads and followed by $OD_{600}$ determination. The

aggregation index (AI) was calculated as the ratio between $OD_{600}$ supernatant/$OD_{600}$ sediment [11]. Control cultures were grown in 7H9 ADS Gly Tw to reduce aggregation. In each case the aggregation assay was performed in triplicate.

**Sliding motility assay.** The assay was carried on by spotting 10 μl of stationary phase culture of each mutant under study in the center of 7H9- 0.4% (w/v) agarose plates with no carbon source added; plates were sealed with Parafilm and incubated at 37˚C for 7 days, at which time visual inspection of plates and measurement of the diameter of the halo of visible growth were carried out.

**Cell envelope analysis of mycobacteriophage resistant *M. smegmatis* mutants.** Extractable lipids were obtained from 100 ml cultures of each mutant under study grown at 150 rpm at 37˚C in 7H9 Gly supplemented with dextrose (2% w/v) until saturation (usually 24 h-36 h) following the protocol described by Chiaradia et al [12]. Cultures were centrifuged at room temperature at 4500 xg and the supernatant discarded; the tubes containing the cell pellet were allowed to dry by inverting the tubes on paper, afterwards kept at 4˚C until its use. Lipid extraction was performed by solvent extraction, sequentially adding methanol/chloroform mixtures at three different ratios (2:1, 1:1 and 1:2 v/v) in volumes equal to the packed cell volume [12]. Briefly, at the time of each addition cells were manually mixed and the extraction proceeded with gentle stirring for 12 h at room temperature; afterwards cell pellets were collected by centrifugation and the supernatants carefully removed. The three supernatants were afterwards combined and dried. TLC analysis of extracted lipids was done on silica gel plates using chloroform/methanol (90:10 v/v) as solvent system [12]. Elimination of the acetyl/acyl substituents of GPLs, yielding the base-stable GPLs, was performed by an alkali treatment as described by Burguiere et al [13]. Visualization of the bands of the different compounds extracted was carried out by spraying the plates with an ethanol solution of α- naphthol followed by the plates heating at 120˚C for 30 min.

## Molecular biology techniques

Chromosomal DNA extraction from *M. smegmatis* mc²155 and derived strains was performed according to Pavelka et al [14]. Amplification by PCR of MSMEG_0392 was carried out using primers MSMEG_0392Fw1 `CCACGTCGTCGTCTTTCAG` and MSMEG_0392Rev `ACGAGTGG CGATGGACGAG` under the following conditions: denaturation (94˚C, 5 min), followed by 30 cycles of denaturation (94˚C, 1 min), annealing (62˚C, 40 sec) and extension (68˚C, 90 sec); the amplification product of the expected size was gel purified and sequenced at a commercial facility.

## Bioinformatics analysis

Genome sequences were analyzed using DNA Master (http://cobamide2.bio.pitt.edu/). Genes were determined using Glimmer11 and GeneMark [15–17] integrated in DNA Master program aided by visual inspection. The presence of tRNA was analyzed by Aragorn [18] and tRNA Scan SE (available at http://cobamide2.bio.pitt.edu). Probable functions were assigned when possible using BLASTP alignments to GenBank and PhageDB database (https:// phagesdb.org/), HHPred [19, 20] and HMMER [21]. ClustalW [22] was used to align Lys A domains while visualization and editing of the alignments were performed using Jalview [23]. Comparative genome analyses of Weirdo19ES were performed with Phamerator [24] using our own database which was generated using the Genebank files from 54 mycobacteriophages belonging to different clusters and subclusters (S1 Table). Pairwise comparisons using ClustalW and BLASTP of gene products made by this program yield families of related sequences which are designated as "phamilies" or phams and numbered as pham 1, pham 2, etc. For our

analysis we considered that two proteins belonged to the same pham if the percent identity and E-value are higher than 32,5% and lower than $1e^{-50}$, respectively. Genome maps of each bacteriophage under comparison were generated by Phamerator; the genes are color coded automatically according to their pham, with "orphams" (those with only a single gene member) shown in white [24]. Genomic parameters (ORF density, gene length, coding percentage and GC content) were obtained through the use of Artemis version 16.0.0 [25].

Genome pair-wise comparison were performed using a Python program, dRep version 2.2.4 [26], this program compares all genomes using Mash [27] and then executes a secondary algorithm (ANImf) on the genomes having more than 90% Mash. ANImf aligns whole genome fragments and filters the alignment, then calculates the nucleotide identity of aligned regions. We set as default an ANI threshold to form primary (MASH) clusters of 90% with a Mash sketch of 50000.

A phylogenetic network of the 55 mycobacteriophages (including Weirdo19ES) was performed using a fragmented alignment approach using Gegenees 3.1 [28] (50 bp fragment size and 25 bp sliding step size), the results were exported as a Nexus file that was loaded into Splitstree 4 software applying a NeighborNet [29] method. CGView comparison tool was used to perform the circular BLAST analysis [30].

The analysis of gene content dissimilarity (GCD) was performed with the Python scripts available in Github (https://github.com/tmavrich/mavrich_hatfull_nature_micro_2017) written by Mavrich and Hatfull [31]. The data used by the scripts to calculate the GCD values were obtained from our Phamerator local database described above. A distance matrix was calculated with the GCD values for each pair of phages using the DendroUPGMA web server (http://genomes.urv.cat/UPGMA/). The Newick tree of the UPGMA clustering was imported with SplitsTree 4, EqualAngle algorithm was used to construct the tree [29].

Relative synonymous codon usage (RSCU) analysis was performed as described by Esposito et al [32]. To this end RSCU values were calculated using MEGA X for each phage analyzed, and the values for stop codons, methionine and tryptophan were removed. Then a similarity matrix of Pearson coefficient correlation was obtained using the DendroUPGMA web server (http://genomes.urv.cat/UPGMA/) [33]. TreeView3 program (https://bitbucket.org/TreeView3Dev/treeview3) was used to perform a hierarchical clustering by Pearson correlation and complete linkage method; the results were represented as a heatmap. All graphics were edited using Inkscape 0.92 (www.inkscape.org). All the analyses were performed on public databases, last accessed on 06/10/2019.

**Nucleotide accession number.** The nucleotide sequence data of the bacteriophage Weirdo19ES has been deposited in GenBank (accession number MN103533).

## Results

### Weirdo19ES genome organization

The genome of Weirdo19ES spans 52581 bps, having a 70.19% GC content with a coding percentage of 96.1% and an average gene length of 568 bps which results in an ORF (open reading frame) density of 1,692 genes per kbp. No tRNA encoding genes were detected by using Aragorn and tRNA Scan SE. The analysis of the genomic sequence of Weirdo19ES predicted 89 ORFs of which more than 50% (48/89) encoded proteins of unknown function. The remaining ORFs (S2 Table) contained readily assignable functions for genes encoding structural proteins of the virion, assembly and lysis proteins, as shown in Fig 1. Interestingly, most of the genes are transcribed in the rightward direction with only a handful of genes (ORFs 33–40, 43, 44, 62 and 77) being transcribed leftward. Clear functions in this group could only be assigned to

**Weirdo19ES**

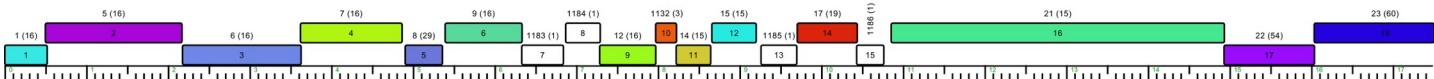

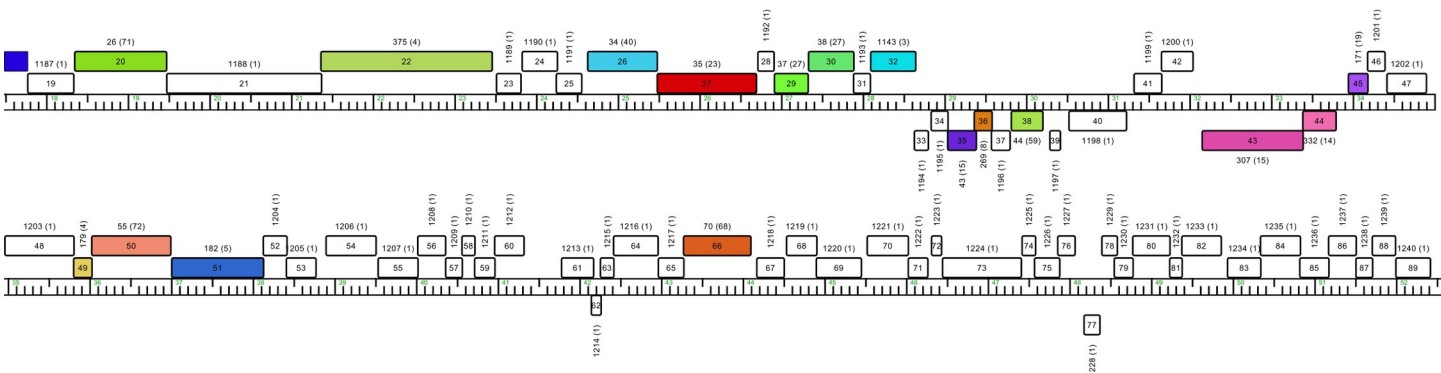

**Fig 1. Weirdo19ES genome map.** This map was generated by Phamerator, the colors of the gene indicate the assigned pham, orphans are shown in white. The pham numbers are above the boxes, whereas the number of the pham members are indicated between brackets.

**A)**

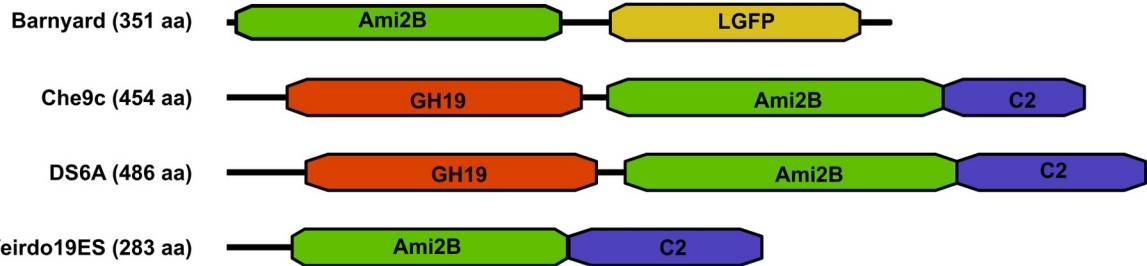

**B)**

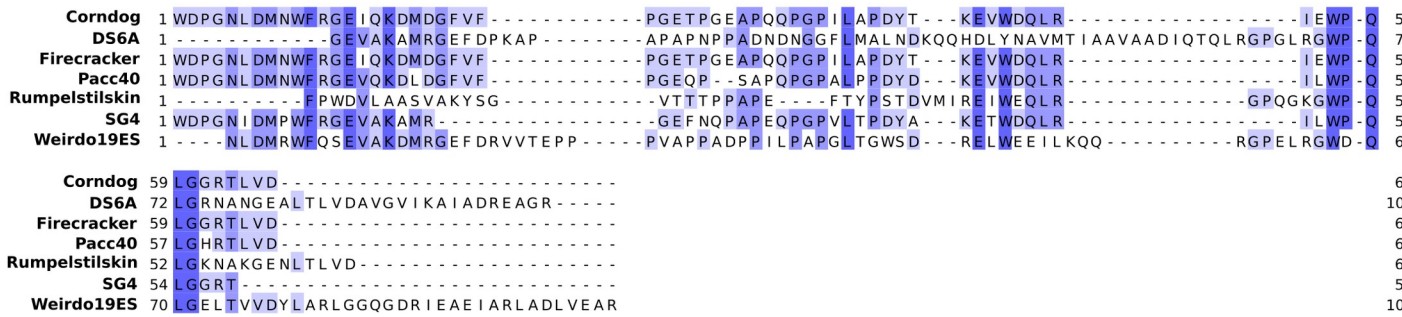

**Fig 2. Schematic representation of the Lys A domains.** A) Examples of the different LysA domain organization as represented by those for mycobacteriophages Barnyard, Wildcat, DS6A, and Weirdo19ES. B) ClustalW alignment was performed with the C2 domains of different reported Lys A enzymes. In this analysis, the C2 domain of DS6A Lys A displayed the highest homology to that of Weirdo19ES (29%). The alignment was visualized and edited by Jalview program [23].

genes 43 and 44 encoding the putative phage integrase (Int) and the immunity repressor respectively (Fig 1).

**Analysis of the left side of the genome of Weirdo19ES.** The analysis of the left side of the genome of Weirdo19ES reveals genes encoding the small and large terminase subunits, followed by ORFs 3–22 encoding recognizable head, tail and assembly proteins including portal, protease, capsid, head-tail connector proteins, major tail subunit, tail assembly chaperones, tape measure protein (TMP), and minor tail protein genes. This organization is shared by many mycobacteriophages, except for ORF 13—with no known function–which is unique to Weirdo19ES.

The predicted lysis functions (ORFs 26–28) are positioned to the right of the structural genes in Weirdo19ES and include an endolysin gene (ORF26, Lys A) a putative mycolyl-arabinogalactan esterase (gene 27, Lys B) and a holin (ORF28) [34]. A detailed review of mycobacteriophage Lys A enzymes shows that there are at least 25 different module organization arrangements in Lys A (named Org-A to Org-Y) based on the presence of one or two enzymatic domains, N-terminal extensions and one of three C-terminal domains (C1-C3) [35]. In most cases, Lys A proteins contain three domains [35]. Of note, Weirdo19ES Lys A is shorter than in the majority of the mycobacteriophages analyzed until now (283 amino acid residues in length), containing only two domains, an ami2B peptidoglycan hydrolyzing domain and a C2 C-terminal domain (Fig 2A). Weirdo19ES lacks a N-terminal domain, a situation also reported in mycobacteriophages of Org-R (WildCat), Org-S (Barnyard) and Org-X (DS6A and Che9d) [35]. Interestingly, Weirdo19ES Lys A displayed good homology to that of the singleton DS6A (E-value $4e^{-99}$), including homology at the C domain. In spite of that, Weirdo19ES Lys A clearly defines a different domain combination.

**Analysis of the central region of the genome of Weirdo19ES.** Genes encoding nonstructural ORFs 33–40. Weirdo19ES genes 33 to 40 are transcribed to the left; although this arrangement of genes encoding proteins of unknown function is conserved in other mycobacteriophages such as mycobacteriophage Giles (ORFs 39–45), the placement is usually at the right of the integration cluster (integrase-repressor-excisionase) while in Weirdo19ES genome this cluster is placed to the left of the integration cassette (Fig 1). Four of the seven leftwards-transcribed genes have no close relatives in other mycobacteriophages, while three (ORF35, 36 and 38) display homology (E-value $e^{-11}$, $e^{-9}$ and $e^{-27}$, respectively) to genes present in phages Webster (which is still under analysis and has not been assigned a cluster yet) and Milly (member of the cluster K2). Milly ORF39 belongs in pham 59124 which also includes homologous ORFs from other 63 phages belonging to different clusters like K, D, O and N. Until now no function as been demonstrated for those genes in spite of their conserved presence in many phages.

**Integration and immunity region.** As for the majority of temperate Siphoviridae mycobacteriophages, the genes playing a role in their integration are located at the middle of the genome; in the case of Weirdo19ES ORFs 43–45 encode a tyrosine integrase, an immunity repressor and an excisionase. ORF43 has homology to members of the pham 43705 and it is related to integrases present in mycobacteriophages of cluster G1 (such as BPs, Halo, Annihilator and Avrafan); it is also closely related to the integrase of phage UmaThurman (pham 41717, having 3 members), a *Gordonia* infecting phage belonging to cluster CV, and a few members of cluster Y [36] (Fig 3A). A representative set of amino acid sequences of repressors (Rep) of temperate phages was used to build a Maximum Likelihood phylogenetic tree, showing that ORF44 of Weirdo19ES is more related to its homologues present in mycobacteriophages of the cluster Y. In this case, the repressors of phages of cluster G1 were more distantly to Weirdo19ES (Fig 3B).

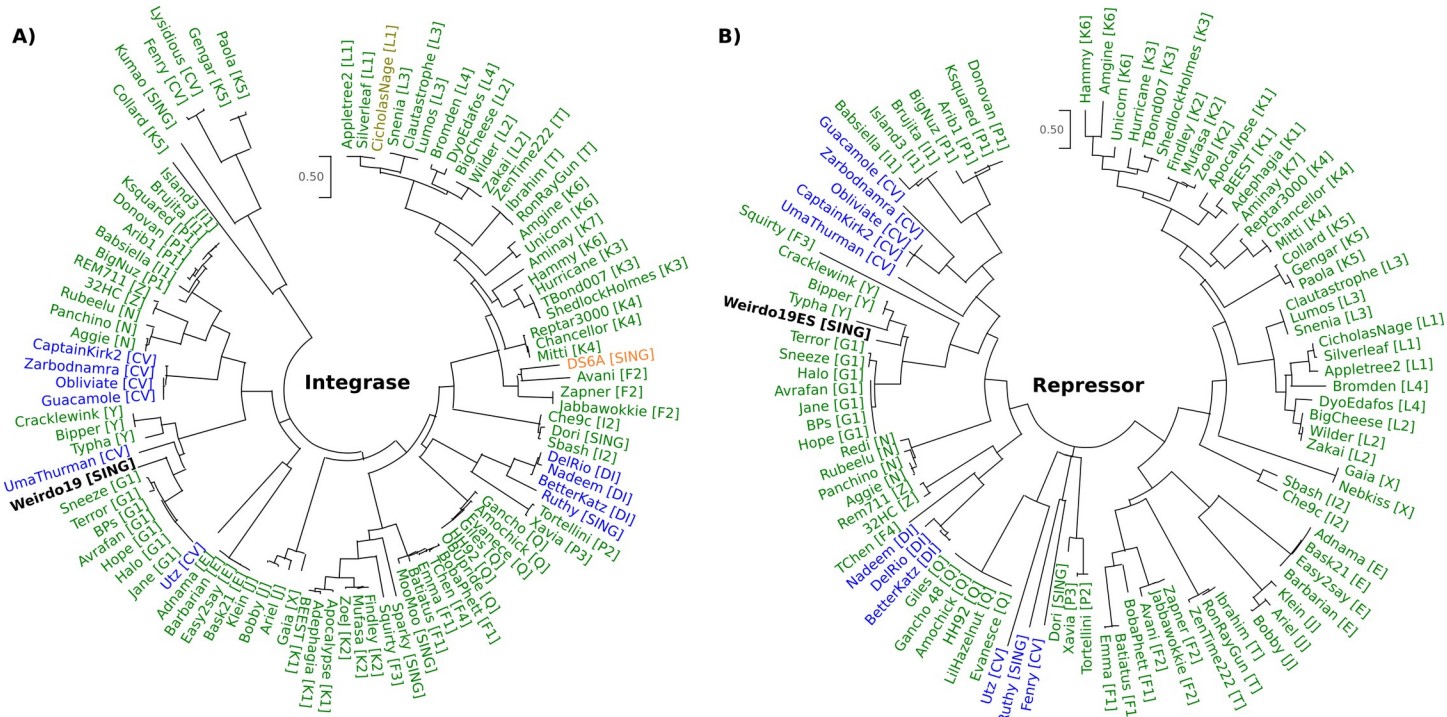

**Fig 3. Phylogenetic analysis of the integrases and repressors.** The multiple sequence alignment of the integrases and repressors were performed using MUSCLE and the evolutionary history was inferred by using Maximum Likelihood method; both analyses were conducted by MEGAX. Circular dendrograms depict the phylogenetic relationships between integrases A), and repressors B). The graphics were colored by host where *M. smegmatis*, *G. terrae* and *M. tuberculosis* phages are shown in green, blue and orange, respectively, and Weirdo19ES is shown in black.

Nucleotide search determined that Weirdo19ES *att*P is encoded in the repressor gene (ORF44), encompassing bp 33321–33354. A similar nucleotide sequence presumably corresponding to the mycobacterial counterpart *att*B, is located at MSMEG_6349, encoding a tRNA-Arg which has already been reported as the attachment site for mycobacteriophage BPs [37]. We found a cognate *att*B in *Gordonia terrae* 3612 genome, also located at a tRNA-Arg, thus suggesting that Weirdo19ES may also be able to lysogenize that species. A peculiar characteristic of the *int-att*P organization is the location of the putative *att*P core which is placed within the coding region of the repressor gene encoded by ORF44, a common trait described for mycobacteriophages Halo and BPs [38]. In agreement with the proposed function, ORF44 contains a pfam01381 motif which is associated with helix–turn–helix DNA-binding motifs and usually present in phage repressors. Finally, ORF45 (a member of the pham 39621 consisting of 54 members) is proposed to be the excisionase protein (Cro) based on its homology (although limited) to several other predicted excision proteins in databases; of those, ORF33 of mycobacteriophage Lemuria (G4 subcluster) is the closest one (e$^{-13}$).

The organization of the integration system of Weirdo19ES is similar to that reported for phages BPs, Halo and Brujita [38], in which Int and Rep are transcribed leftward and Cro rightward. The intergenic region between Rep and Cro proteins (151 bp), contains a palindromic region that could function as a regulatory motif for the expression of these genes (Fig 4A). The mechanism of integration-dependent bacteriophage immunity, which was determined in detail for the mycobacteriophages mentioned above, is based on the stability of the integrase and repressor proteins which depends in turn on the action of a host protease (ClpXP) on proteins containing a specific aminoacid tag (*ssr*A) [38]. The alignment of the C-

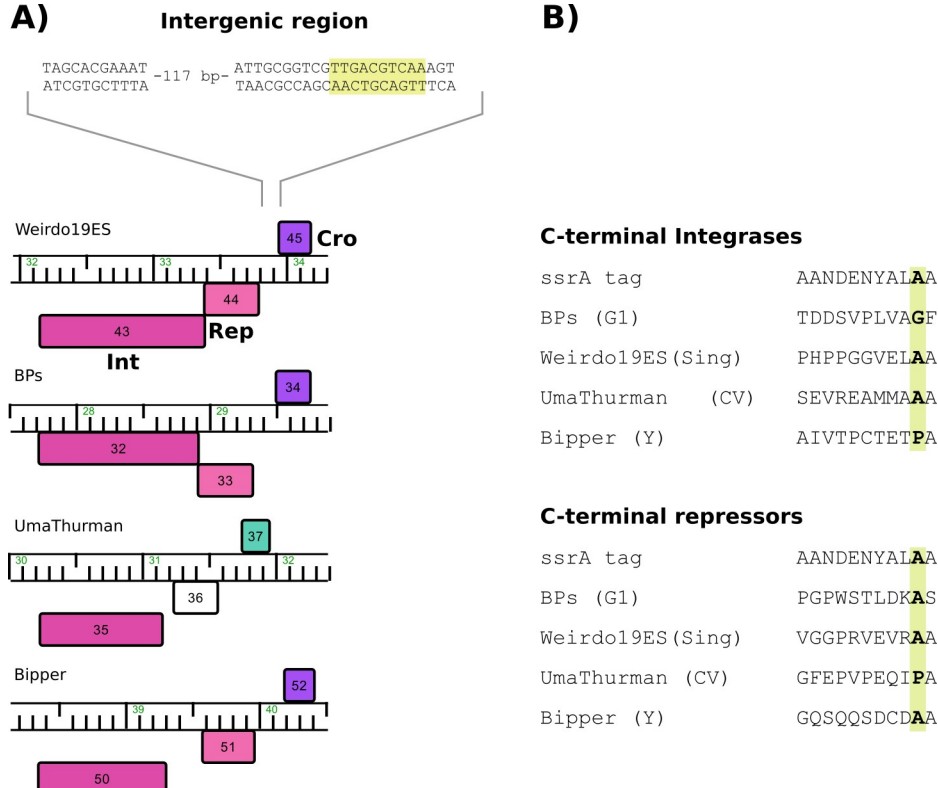

**Fig 4. Analysis of the integration system.** A) Graphical representation of the organization of integration system. The integration module of the Weirdo19ES, BPs, UmaThurman and Bipper were extracted from Phamerator, where the color indicated the assigned pham. Rep protein from UmaThurman phage could not be assigned to any pham (white box). The intergenic region between *rep* and *cro* genes was analyzed and a palindromic sequence was identified as indicated by a yellow box. B) The C-terminal sequence of the repressors and integrases were aligned with the *ssr*A proteolytic tag of the *M. smegmatis*, where the conserved alanine residue is colored.

termini of these proteins and the *ssr*A tag of *M. smegmatis* showed a conserved Ala residue in Weirdo19ES, which is known as critical for the action of the protease on the integrases analyzed (Fig 4B). Intriguingly the C-terminal domain of the integrases of phage BPs has a Gly residue and that of phage Bipper has a Pro residue instead of the conserved Ala while the C-term domain of the repressor of phage UmaThurman has a Pro residue instead of the Ala residue (Fig 4B). In all the cases analyzed a second Ala residue is adjacent to the first one (with the exception of phage BPs which contains a Ser residue) (Fig 4B), thus mutational analysis will be required to identify which residue is the critical one.

**Analysis of the right side of the genome of Weirdo19ES.**   Finally, the right side of Weirdo19ES contains non-structural genes of which only a few could be assigned function, among them ORF66 (Ruv like resolvase) and ORF67 (WhiB-like transcriptional regulator). ORF86 is only present in the two phages forming the Z cluster, REM711 and 32HC, and in both cases the homology was low.

**Free standing HNH endonucleases.**   The genome of Weirdo19ES contains an ORF (ORF89) which encodes a protein with a length of 114 amino acids in which a HNH endonuclease domain was predicted by InterProScan. A BlastP search of the Actinobacteriophage database (available in www.phagesdb.org) using this putative HNH shows that it is similar to gp54 of the actinophage KaiHaiDragon (E-value $4e^{-16}$), which infects *Microbacterium foliorum*. This protein is member of the pham 36880, having 12 members. ORF61 is also

proposed to be HNH; using HMMER it was recognized as a member of the HNHc_6 family (PF16784.59), although showing a low similarity to the family domain.

## Comparative genomics of Weirdo19ES

In order to gain insight on the relationship of Weirdo19ES to other mycobacteriophages we first compared its genome using BLASTN against the Actinobacteriophage database. Phages belonging into clusters G, K, CV and DI (this last two containing *Gordonia* phages) showed homology to Weirdo19ES. Interestingly, the most significant alignments to Weirdo19ES were those of two singletons; Ruthy which infects *Gordonia spp*. and DS6A which is the only mycobacteriophage directly isolated using *M. tuberculosis* as host (E-value of $2e^{-52}$ and $6e^{-47}$, respectively). Further comparison was performed against a set of 54 phages (chosen randomly from each cluster and subcluster of phages displaying the highest homology to Weirdo19ES) belonging to mycobacteriophage subclusters G1-4, Q, K2, X, Z, 5 mycobacteriophage singletons and 17 phages that infect *Gordonia spp*. belonging into the clusters CV and DI. The results represented in CGView BLASTN show that there is little sequence homology between Weirdo19ES and the 54 phages under comparison (S1A Fig).

Interestingly, although displaying a very low nucleotide sequence similarity, the genes present in Weirdo19ES have homologues to phages belonging to cluster Q (11%), Z and G (8% each one). However, when the comparison was made on the basis of the encoded proteins, we detected higher homology between the phages on proteins encoded on the left half of their genomes. This result is expectable as structural proteins are encoded in that part of the phages genomes; each of those proteins very frequently have similar folds and domains dictated by conserved structures of virion particle components. Nevertheless, the proteins encoded on the right half of the genomes are much less homologous or completely unrelated (S1B Fig).

A phylogenetic network of the 54 phages mentioned above plus Weirdo19ES was built using Gegenees and Splitstree; the results obtained again located Weirdo19ES as a unique phage in a separated branch (Fig 5). A distant relationship with phages of cluster Q was observed when the area of the center of the network was enlarged and analyzed; mycobacteriophages DS6A (a singleton) and the two members of cluster Z (REM711 and 32HC) were the closest ones to Weirdo19ES (Fig 5).

A pair-wise comparison of all genome was performed using dRep [26], and a primary cluster was made using Mash, with ANImf used to perform a secondary, more sensitive clustering (Fig 6). This analysis also showed that Weirdo19ES is a singleton with a distant relation to bacteriophages members of the cluster Q. To confirm these results, we compared the sequence of TMPs, conserved proteins that have been used to cluster mycobacteriophages [39]; again our results showed that the Weirdo19ES TMP was close to homologues belonging to cluster Q (S2 Fig).

Finally, we sought to determine whether Weirdo19ES was originally hosted by *Gordonia*, switching genus barriers and beginning to infect a yet undetermined mycobacterial species by the time we isolated it. In order to accomplish that goal, we first utilized a novel measure recently developed by Hatfull´s group, termed GCD which has been applied to study how *Gordonia* phages are related to other actinobacteriophages by comparing phams [4]. GCD correlates shared and unshared phams, and its values range from 1 (no genes are shared) to 0 (identical gene content). Comparison of Weirdo19ES versus representatives of each cluster and subcluster under study and singleton mycobacteriophages was carried out. In this way, we obtained a representation in which mycobacteriophages from cluster Q and Z were the most similar to Weirdo19ES (average GCD = 0.74 and 0.804 respectively), while phages infecting *Gordonia* and belonging into cluster DI had a GCD value of 0.87 (Fig 7A). Thus this analysis

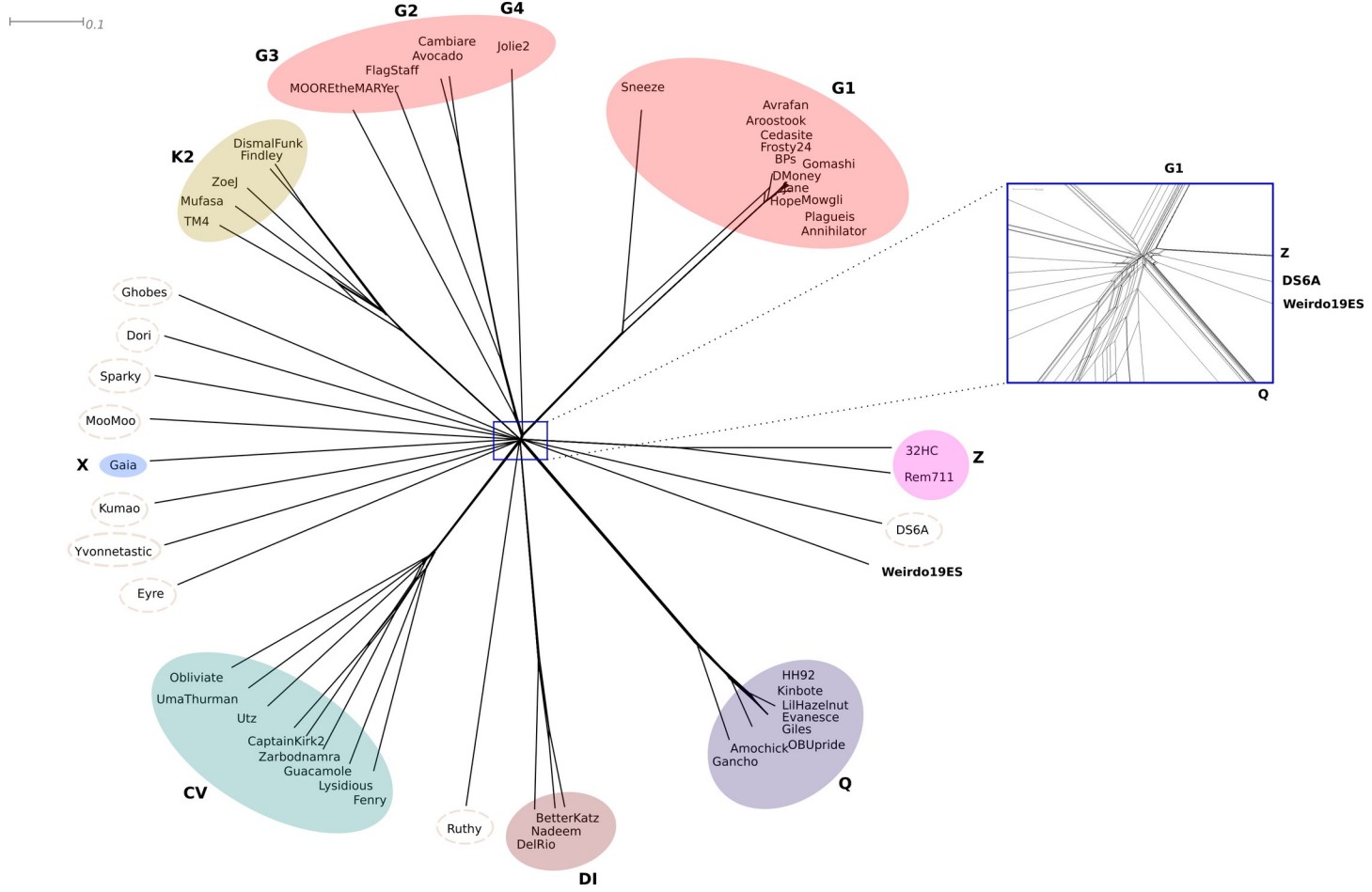

**Fig 5. Phylogenetic network of mycobacteriophages.** Gegenees was used to calculate the distance between bacteriophages and the phylogenetic network was build using the Neighbor-Net algorithm in Splitstree 4. The different clusters of bacteriophages were represented in different colors. The center of the network was zoomed in to analyze the most distant relationships.

did not clarify whether Weirdo19ES was originally a *Gordonia* infecting phage as the mycobacteriophages under analysis displayed GCD values larger or smaller than those displayed by phages infecting *Gordonia*. Importantly all the singleton mycobacteriophages displayed the highest level of dissimilarity indicating that the analysis was valid (Fig 7A, for the sake of clarity, a partial scale of GCD values is shown). When all the phages including Weirdo19ES were compared against each other, the results also showed that phages from cluster Q were the most similar ones as assessed by GCD values. Interestingly, Weirdo19ES and phages of cluster Q and Z were more related to *Gordonia* phages like Ruthy, Ghobes and members of clusters CV and DI, than to mycobacteriophages of clusters G, K2 or X (Fig 7B).

A different approach to assess the role of *Gordonia* as a host for Weirdo19ES was undertaken by analyzing the RSCU which measures the relative frequency of codon usage in bacterial hosts and bacteriophages [8]. As shown in Fig 8, in the resulting dendrogram (left side of the heatmap) all phages analyzed are assigned to four different clusters (1 to 4) as defined by RSCU. Weirdo19ES is a member of cluster 3, in close relation with the cluster K2 mycobacteriophages ZoeJ and Dismalfunk. Bacteriophages members of the cluster DI of *Gordonia* as well as a singleton were positioned on a separate branch on the same cluster. Interestingly, two phages previously isolated by our group (Jolie2 and 32HC, belonging into cluster G4 and

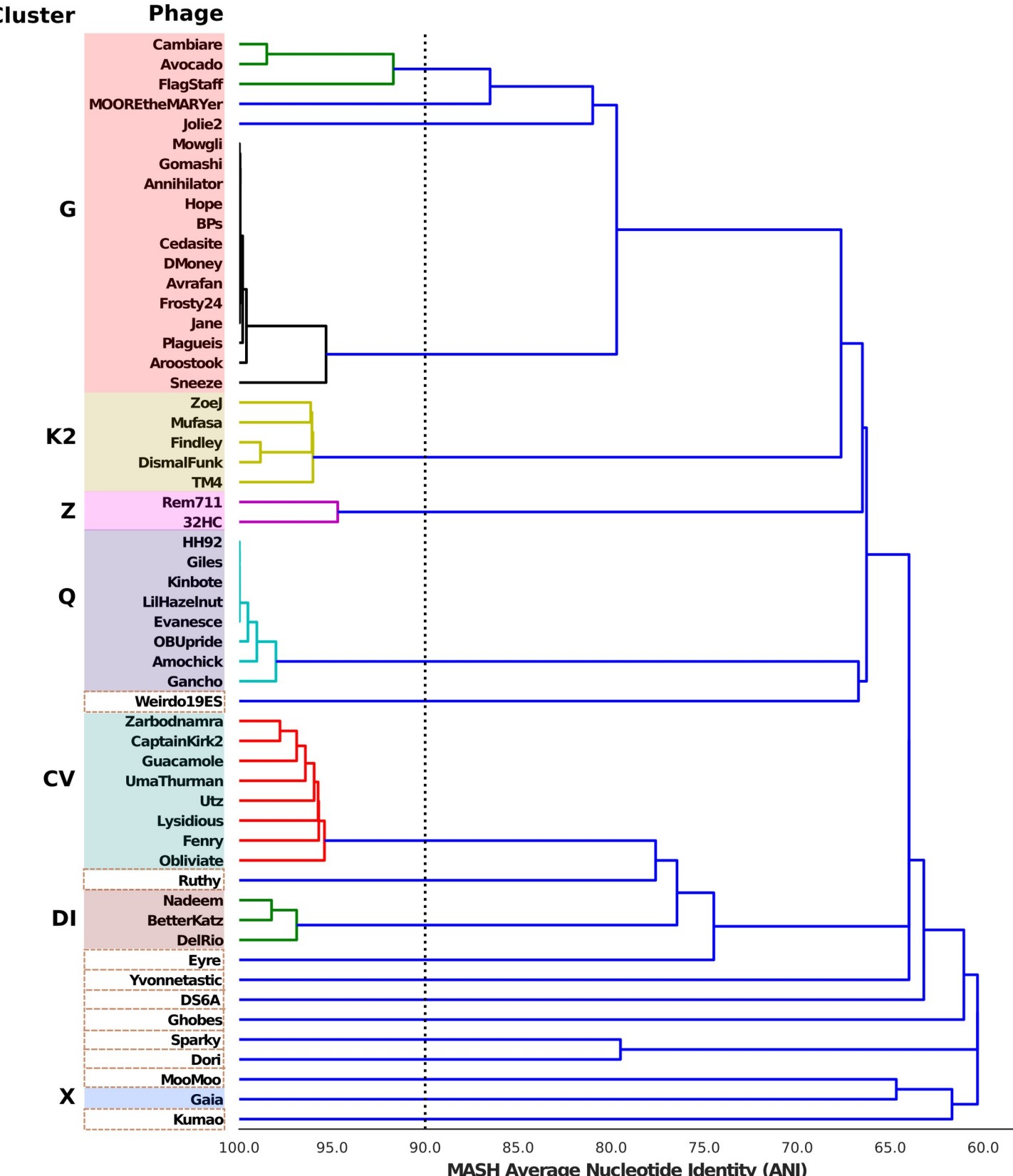

**Fig 6. Mash distance dendrogram.** Primary clustering was performed using the pair-wise Mash distance with a 90% of ANI (dotted line) in dRep program [22]. The different mycobacteriophages clusters were colored and singletons are shown in dotted boxes.

A)

B)

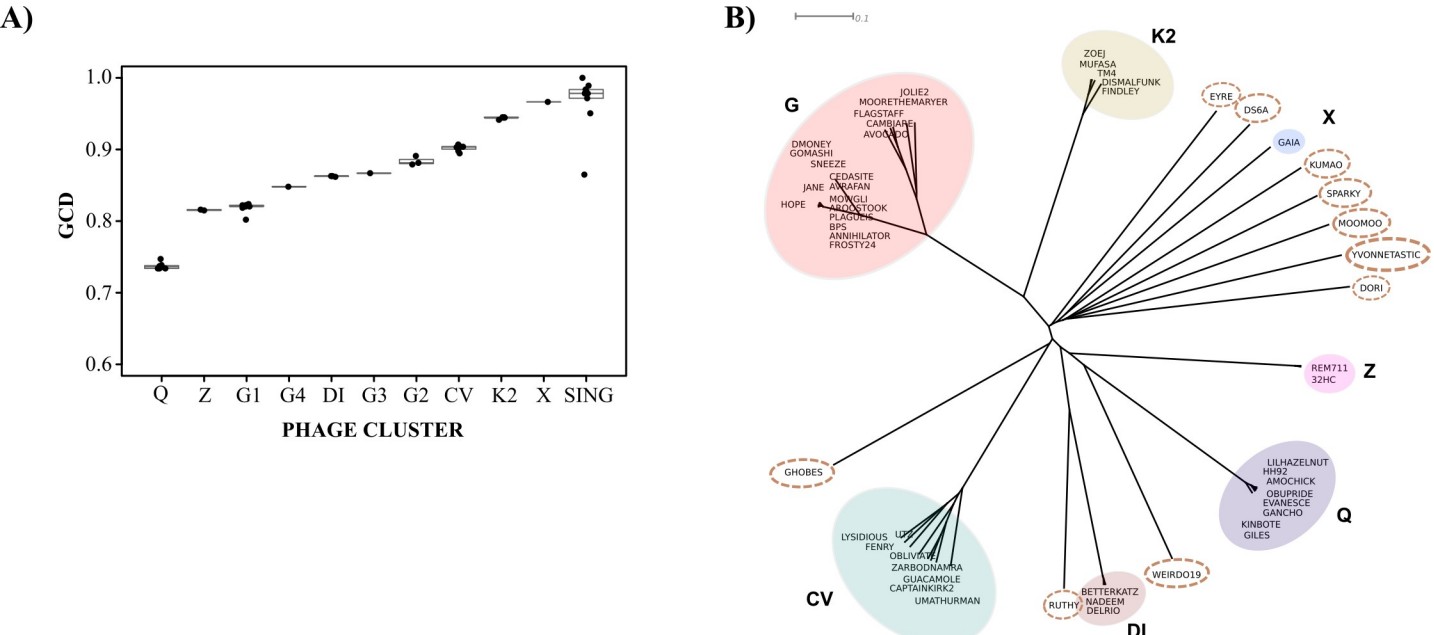

**Fig 7. Gene content dissimilarity (GCD) analysis.** A) Boxplot of the GCD values of 54 bacteriophages against Weirdo19ES. As shown in picture, cluster Q are sharing more genes with Weirdo19ES, represented by the lower values of GCD. B) EqualAngle algorithm was used to build a phylogenetic network in Splitstree 4 with the GCD values of all bacteriophages in the analysis. The different clusters were colored whereas singletons are in dotted boxes.

cluster Z respectively), also clustered with Weirdo19ES, in spite of having phylogenetically distant genomes.

## Biology cycle of Weirdo19ES

To add information of mycobacteriophage Weirdo19ES, we determined some of its biological features such as infection cycle parameters and phage-bacterial interactions (see Materials and methods). Our analysis showed that Weirdo19ES was able to lysogenize the mycobacterial host at a frequency of 12% (an average of three independent experiments) which is intermediate between the frequency displayed by L5 (22%) or Giles (2%) as previously reported [2]. This singleton also had an unusual lytic cycle characterized by an extremely long latency time of ≈300 min which largely exceeds those reported for other mycobacteriophages such as Kampy which showed a latency time of 100 min [40] or D29 (100–120 min) [41, 42].

There are very few reports on the biology behind *M. smegmatis* resistance to mycobacteriophages or the nature of receptors for mycobacteriophages, being the mutants resistant to the transducing phage I3 the best characterized till now [43, 44]. In order to gain knowledge on the subject we obtained several spontaneous *M. smegmatis* mutants resistant to Weirdo19ES (see Materials and methods), one of which, designated 1R-Weirdo19ES, was kept for further analysis. Simultaneously we isolated spontaneous *M. smegmatis* mutants resistant to mycobacteriophage I3 following the same protocols, keeping one clone which we designated as 3R-I3. Importantly, successive passages on fresh medium showed that morphotypes were stable although revertants with colony morphology similar to the parental strain arose with a frequency of $10^{-6}$–$10^{-7}$. In order to characterize those phage resistant mutants, aliquots of each one of them, along with the parental strain were plated onto solid medium in the presence of the hydrophobic dye Congo Red; observation of the plates at low magnification revealed that I3 and Weirdo19ES *M. smegmatis* resistant mutants displayed a rough colony phenotype being

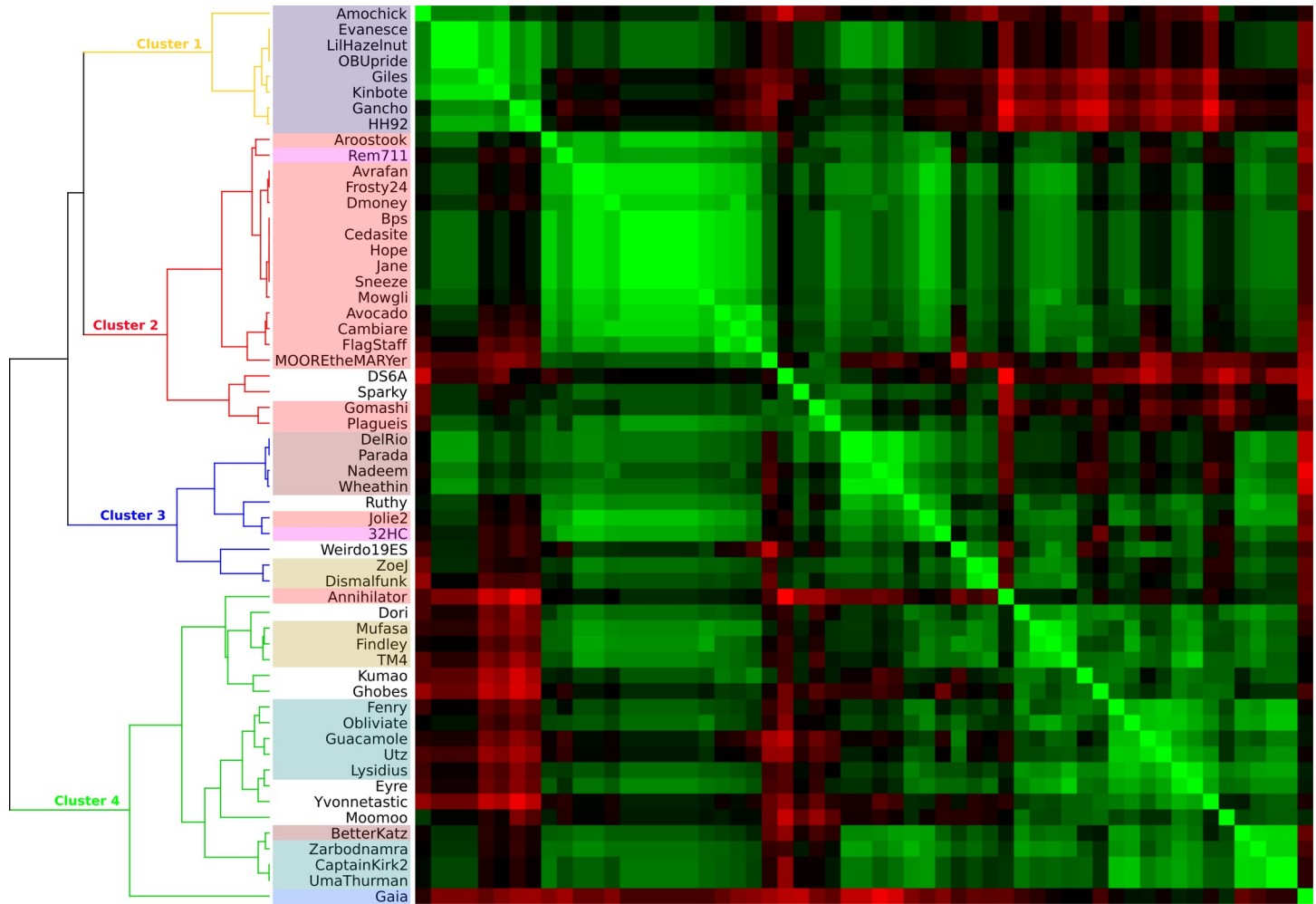

**Fig 8. Clustering analysis of the relative synonymous codon usage values.** RSCU values obtained by MEGA X where used to construct a similarity matrix. Hierarchical clustering and heatmap representation was performed in TreeView3 program. The four main clusters are depicted in the dendrogram at the left of the heatmap.

approximately half the size of that of the wild-type *M. smegmatis* mc$^2$155 (Fig 9 panel A). *M. smegmatis* Myc55, a Tn611 transposon insertional mutant affecting GPL synthesis [45] that was added as control, also displayed a highly similar colony morphology in agreement with what was described in the literature (Fig 9 Panel A). The behavior of the phage resistant mutants on sliding motility was also tested finding that while the parental strain displayed the well-known growth pattern the three mutants (1R-Weirdo19ES, 3R-I3 and Myc55) failed to move from the point where the cultures were spotted onto the plate (Fig 9 Panel B). In concordance with the loss of the sliding motility, the aggregation index showed that the three mutants were much more prone to aggregation (4–5 fold) than the wild-type strain (Fig 9).

Determination of the efficiency of plating of each mycobacteriophage against the corresponding resistant mutant revealed a decrease in the efficiency of plating of 10$^5$-fold or more confirming the resistance phenotype (Fig 10). Importantly, I3 failed to propagate normally in *M. smegmatis* mutant 1R-Weirdo19ES; conversely Weirdo19ES propagated at very low efficiency of plating in 3R-I3 (Fig 9). A similar behavior was displayed by *M. smegmatis* Myc55 and thus the three strains showed cross-resistance to both Weirdo19ES and I3; opposite to

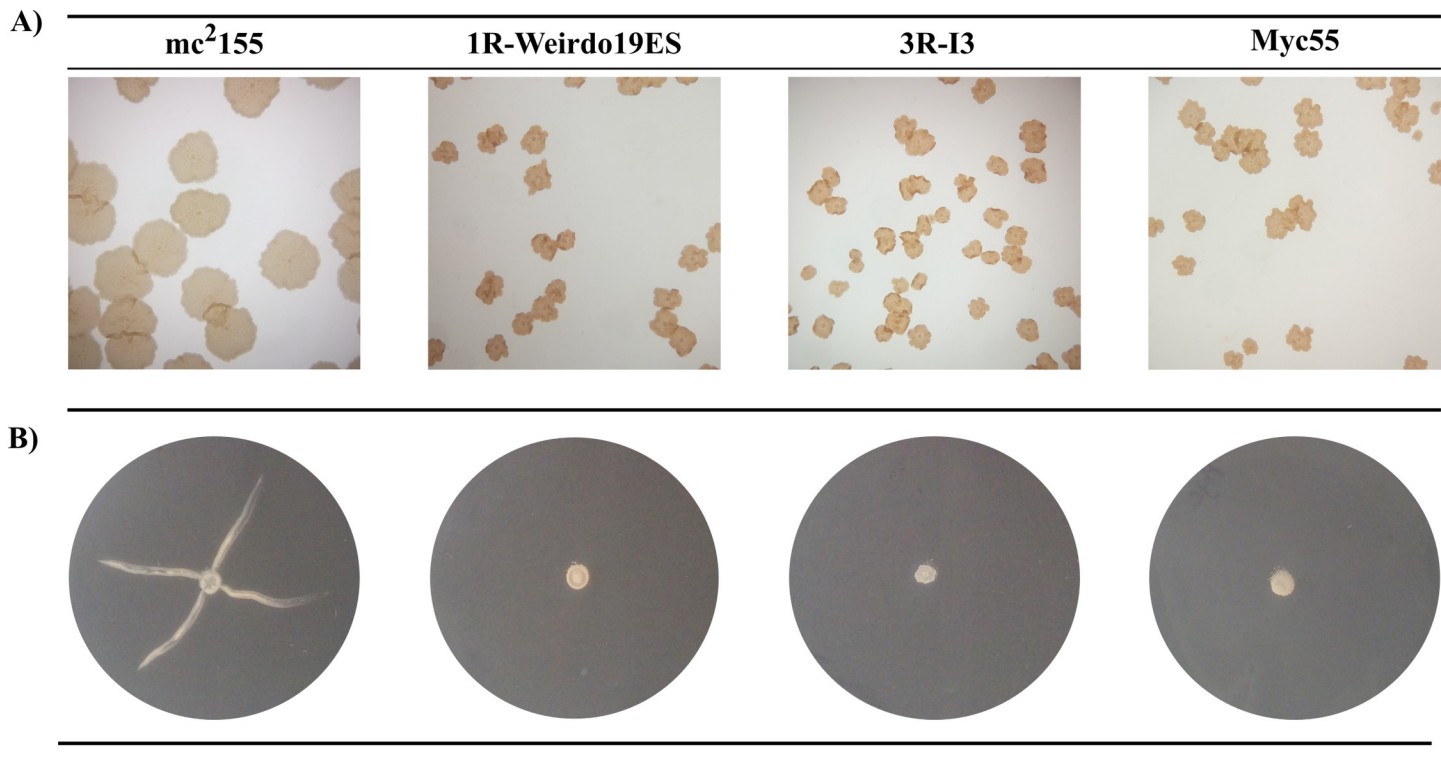

**Fig 9. Phage resistant mutants of *Mycobacterium smegmatis*.** The mutant strains (1R-Weirdo19ES, resistant to mycobacteriophage Weirdo19ES; 3R-I3, resistant to mycobacteriophage I3 or Myc 55 share colony morphology (Panel A), sliding motility behavior (Panel B) and cell aggregation (Panel C). Panel A: Aliquots containing $10^2$ CFU of each strain were spread onto solid medium plates containing Congo Red (100 μg ml-1). Growth was examined under microscope at 8X magnification after 3 days of incubation at 37˚C. Pictures are representative of ten fields observed for each plate. Panel B: 5μl drops of a stationary culture of each strain were placed in the center of 0.3% (w/v) agar plates, the plates were sealed with Parafilm and incubated for 7 days at 37˚C. Panel C: Aggregation Index for each strain was determined in the absence or presence of 0.5% (w/w) Tween 80 as described in Materials and Methods.

that, D29 -used as a control phage- propagated with no substantial differences in the wild-type and phage resistant strains (Fig 10).

Colony morphology of 1R-Weirdo19ES and 3R-I3 *M. smegmatis* resistant mutants suggested cell envelope alterations, hypothesis supported by the colony size and morphology shared by these two mutants and Myc55. A previous report on transposon insertions causing resistance to I3 and yielding a rough colony phenotype and loss of GPL synthesis led us to extract and analyze cell envelope lipids of 1R-Weirdo19ES as described in Materials and Methods [12, 45]. Extraction and chromatographic analysis of lipids of the cell envelope of *M. smegmatis* mc²155, 1R-Weirdo19ES, 3R-I3 and Myc55 supported our hypothesis as no polar GPLs were discernible by TLC analysis in the three mutants while the parental strain displayed the normal content of polar and apolar lipids (Fig 11). Using Tn5730 insertional mutagenesis

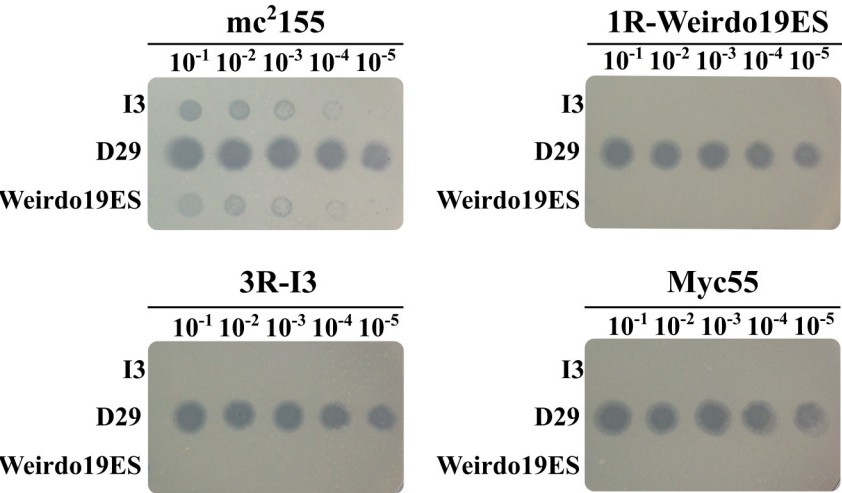

**Fig 10. Cross-resistance of mutants 1R-Weirdo19ES, 3R-I3 and Myc55 to mycobacteriophages Weirdo19ES and I3.** Aliquots (5 μl) of ten-fold dilutions of Weirdo19ES, I3 or D29 (used as control) were spotted on Indicator Plates containing either the wild-type strain, 1R-19WeirdoES, 3R-I3 or Myc 55 on solid medium. Lysis was determined after 24–48 h of incubation at 30˚C.

Chen and co-workers isolated *M. smegmatis* I3 resistant mutants which upon analysis showed insertions in four genes [43]. We selected one of them, MSMEG_0392, encoding the

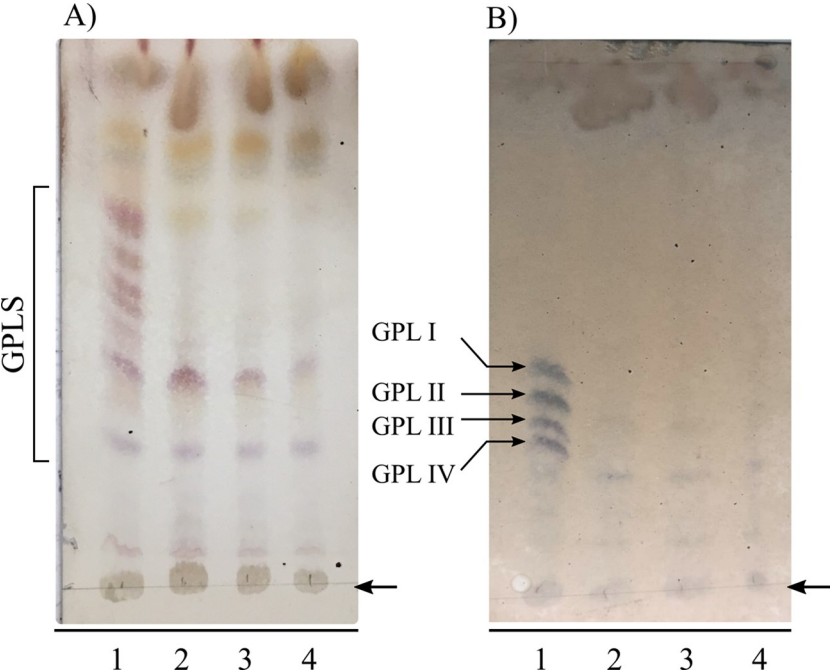

**Fig 11. Mutants resistant to mycobacteriophages I3 or 19WeirdoES lack polar GPLs.** Extractable lipids from strains *M. smegmatis* mc²155, 1R –Weirdo19ES, 3R-I3 and Myc55 were loaded and separated by HPLTC using chloroform: methanol (9:1 v/v) as mobile phase followed by detection using α-naphtol and charring (Panel A). The extracted lipids were also deacylated by mild alkali treatment as described by Burguiere et al. [13], separated on HPTLC plates using chloroform/methanol/water (90:10:1 v/v) and revealed as above; the major GLP species (1–4) are indicated (Panel B). In both panels the origin of the solvent run is shown by an arrow, lanes 1 through 4 indicate the lipids extracted from *M. smegmatis* mc²155, 1R-19WeirdoES, 3R-I3 and Myc 55 respectively.

glycosyltransferase *gtf*, as an initial candidate possibly underlying the phenotype of 1R-Weirdo19ES. PCR amplification and sequencing of this gene showed a single bp insertion at position 575 of the gene, generating a frameshift that created a premature protein termination. Thus, although the involvement of other genes was not addressed (see Discussion), MSMEG_0392 is affected in independent mutants resistant to phages I3 or Weirdo19ES, giving support to our hypothesis and being consistent with GPL analysis.

In conclusion, Weirdo19ES uses one or more of the *M. smegmatis* GPLs as a receptor, as proven by the defect in GPL synthesis in mutants resistant to this phage.

## Discussion

### Weirdo19ES is a singleton mycobacteriophage

There are currently over 1,700 mycobacteriophage genomes that have been sequenced and analyzed (information compiled in www.phagesdb.org), puzzlingly, among those phages, only ten are singletons according to the latest actinobacteriophage database release (last accessed Nov. 1, 2019). We report herein the characterization of another singleton, Weirdo19ES, our first one among a small handful (less than 30) of phages isolated and studied in Argentina [5, 6]. Of note, Weirdo19ES was isolated in 2009 and since then still remains a singleton in spite of the large number of mycobacteriophages that were sequenced since then. This mycobacteriophage has very low nucleotide and protein homology to other actinobacteriophages as judged by BLASTN and BLASTP and graphically depicted in S1 Fig. The analysis of GCD showed that all the singleton phages reported and of Weirdo19ES have very little content of common genes when compared to mycobacterial and *Gordonia* phages in clusters, moreover most of the singletons differ in gene content among themselves (Fig 7A and 7B).

Weirdo19ES has several very interesting features when a specific protein or set of proteins is compared to the homologues present in other actinobacteriophages, one of such features is TMP, the protein that controls the length of the phage tail. Weirdo19ES TMP is related to TMPs of several mycobacteriophages belonging to cluster Q -of which mycobacteriophage Giles is the one more thoroughly studied [2, 46, 47], mycobacteriophage Gaia (one of the two currently known cluster X mycobacteriophages), the singletons MooMoo and Eyre, and cluster Z mycobacteriophages Rem711 and 32HC. However, while all the cluster Q phages have very similar TMPs suggesting a close evolution, Weirdo19ES, as well as the remaining phages mentioned above, is more distantly related. This is more compelling when looking at TMPs from phages Weirdo19ES, Gaia and MooMoo (S2 Fig). Due to that relatedness to singletons or phages infecting *G. terrae* it is possible that Weirdo19ES has been circulating between *Gordonia* and mycobacteria although other hosts cannot be excluded.

Recently, Payne and Hatfull analyzed 224 Lys A enzymes encoded by sequenced mycobacteriophages, pointing out the remarkable diversity of those enzymes both in size and in the encoded activities [35]. Most of the Lys A enzymes analyzed in that report contain three conserved domains, namely a C-terminal domain that is possibly involved in the binding to the mycobacterial cell wall, a central domain that contains a sequence motif associated with peptidoglycan hydrolysis, and a peptidase encoding N-terminal domain [34]. The central domain typically consists of amidases (Ami), muramidases or transglycosylhydrolases (GH) which can cleave different linkages in the peptidoglycan structure [35]. The domains encoding the amidases can be grouped in two families, Ami-2A and Ami-2B according to sequence similarities [35]. Importantly, only the singletons Weirdo19ES, DS6A and the V cluster phage Wildcat lack the N terminal domain encoded peptidase. In both DS6A and Wildcat Ami-2B and GH19 domains are present however Weirdo19ES contains only a central Ami2-B domain (Fig 2). Thus, although several combinations of domains are present in the mycobacteriophage

endolysins analyzed by Payne and Hatfull [35], Lys A from Weirdo19ES is still an oddity. There are very few reports on the specific activity of Lys A in mycobacteriophages [42], thus, considering the extremely long latency time (300 min) displayed by Weirdo19ES, a possibility worth exploring is its correlation to a less than optimal Lys A activity.

A set of 8 leftward transcribed genes (ORFs 33 to 40) is also of interest, in first place in the cases of phages where such a set is found -such as mycobacteriophage Giles [48], it is located at the right side of the integration cassette while in Weirdo19ES it is to the left as is also located in phage Tweety. In second place, in the case of Giles, the set is formed by ten genes transcribed leftward, spanning form ORF38 to ORF47: of those 7 are "orphams". The analysis of essentiality of those genes in phage Giles indicated that only ORF40 and ORF42 encoded essential proteins that were also strictly required for obtaining lysogens while ORF41, although not essential, reduced lysogenization in 70% [2]. Importantly deletion of ORF47 (Cro repressor) reduced the number of phages per plaque as well as abrogated the lysogenization. RNA-seq studies showed a low level of expression of the leftward transcribed genes (with the exception of ORF47 during the lysogenic state) regardless of which phage replication stage (early, late or lysogeny) was analyzed [2]. Thus although we cannot assign roles to the Weirdo19ES leftward transcribed set of genes, it is reasonable to postulate that they may perform roles similar to those of phage Giles.

Temperate bacteriophages encode transcription repressors required to maintain lysogeny and confer superinfection immunity, thus regulation of those repressors is crucial for the choice of lysogenic or lytic growth. Temperate mycobacteriophages contain two types of integrases classified depending on their active site, of which tyrosine integrases (Y-Int) are favored against Serine integrases (S-Int). In recent years a new subclass of Y- Ints has been described which lacks an N-terminal arm-type binding domain [38]; a remarkable feature of those systems is that the *att*P site is located within the gene encoding the phage repressor, such that integration results in loss of the 3′ end of the gene and removal of a C-terminal degradation tag that would otherwise inactivate the virally-encoded form of the repressor [38]. Mycobacteriophage Brujita, that contains such a Y-Int enzyme, has recently been characterized in detail as a simple system in which the rate of integration and excision could be dictated by Int instability or concentration [49]. The lack of N-terminal Int domains, arm-type sites, and recombination directionality factors make the integration system of Brujita very simple [49]. In a similar way, our analysis of mycobacteriophage Weirdo19ES revealed the presence of a Y- Int encoding gene (ORF43) with homology to that of phage Brujita (Fig 3); however, a detailed inspection of the relationships between the Y-Int available in the databases indicates that Weirdo19ES Int is related to integrases of phages belonging to cluster G as well as to the one present in Gordonia phage UmaThurman (Fig 3), all of which are included in pham 41717, while Brujita´s Int belongs to pham 44980.

## M. smegmatis resistance to Weirdo19ES highlights the role of gycopeptidolipids as a key mycobacteriophage receptor

Early work from different groups contributed information on the chemical nature of possible mycobacteriophage receptors extracted by solvent treatment of mycobacterial cells; this fraction was composed of amino acids, sugars modified by oxalic acid or pyruvic acid and fatty acids and was capable of interfering with the adsorption of a phage on *Mycobacterium phlei* [50]. Thus, it was clear on the basis of the interference of bacteriophage adsorption to mycobacterial cells that one or more GPL species are able to act as part of the bacteriophage receptor. In spite of that, it was only in 2009 that specific genes were linked to resistance to mycobacteriophage I3 through transposon insertions leading to the lack of production of

GPLs [43]. The insertions yielded four I3 resistant clones and were mapped into genes MSMEG_0392 and MSMEG_0399-MSMEG_0401 encoding a glycosyltransferase (*gtf*2), a MbtH-like protein, and two Non-ribosomal peptide synthases 1 and 2 (*mps*1 and *mps*2) respectively. Although colony morphology alterations were not reported by the authors, Myc55, carrying a Tn611 insertion in a polyketide synthase gene (MSMEG_0398, *pks*) showed the same rough colony morphology and loss of GPLs [45]. In our hands all the isolated *M. smegmatis* clones spontaneously resistant to Weirdo19ES displayed the same rough colony morphology as strain Myc55 and *M. smegmatis* I3 resistant mutants; additionally, the three strains displayed cross- resistance to I3 and Weirdo19ES. Moreover, the genetic lesion in 1R-Weirdo19ES affected MSMEG_0392, in agreement with its loss leading to resistance to I3; finally the GPL profile on TLC plates correlated with those genetic defects.

As a whole, our isolation genome annotation and biological characterization of the single-ton mycobacteriophage Weirdo19ES adds new information to the field and opens up new possibilities of research on the role of the mycobacterial GPLs as receptors for mycobacteriophages.

## Supporting information

**S1 Fig. Comparative analysis using CGView comparison tools.** Circular maps comparing Weirdo19ES with a set of bacteriophages of different clusters infecting either *Gordonia* or *Mycobacterium* hosts are displayed. The outermost ring corresponds to the reference genome and the internal rings shows the regions of BLASTN and BLASTP homology, A and B, respectively. The inner ring displays the GC content of Weirdo19ES.
(TIF)

**S2 Fig. Phylogenetic analysis of the tape measure proteins.** The amino acid sequences of the TMPs were aligned with MUSCLE and the dendrogram was constructed using Maximum Likelihood method in MEGAX: mycobacteriophage clusters are indicated in colored boxes.
(TIF)

**S1 Table. List of mycobacteriophages used throughout this study.** Corresponding accession numbers, host and clusters classification are listed.
(DOCX)

**S2 Table. List of predicted gene of mycobacteriophage Weirdo19ES.** The function were predicted by BLASTP analysis using phagedb database. [1] The highest homologue to each gene product are listed together with the phage cluster and BLASTP E-value [2].
(DOCX)

## Acknowledgments

We gratefully acknowledge the assistance of Dr. Enrique Morales in microscopy and imaging. We also acknowledge the use of the Microscopy facility at IBR (Instituto de Biología Molecular y Celular de Rosario), Argentina. HRM acknowledges Mr. A. Fornasari for secretarial assistance.

## Author Contributions

**Conceptualization:** Héctor Ricardo Morbidoni.

**Data curation:** Cristian Alejandro Suarez, Jorgelina Judith Franceschelli.

**Formal analysis:** Cristian Alejandro Suarez.

**Funding acquisition:** Héctor Ricardo Morbidoni.

**Investigation:** Jorgelina Judith Franceschelli, Sabrina Emilse Tasselli, Héctor Ricardo Morbidoni.

**Methodology:** Héctor Ricardo Morbidoni.

**Project administration:** Héctor Ricardo Morbidoni.

**Writing – original draft:** Cristian Alejandro Suarez, Héctor Ricardo Morbidoni.

**Writing – review & editing:** Héctor Ricardo Morbidoni.

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
