## [Decision Letter · Decision Letter 0]

16 Jan 2020

PONE-D-19-31407

Weirdo19ES is a novel singleton mycobacteriophage that selects for glycolipid deficient phage resistant Mycobacterium smegmatis mutants

PLOS ONE

Dear Dr Morbidoni,

Thank you for submitting your manuscript to PLOS ONE. After careful consideration, we feel that it has merit but does not fully meet PLOS ONE’s publication criteria as it currently stands. Therefore, we invite you to submit a revised version of the manuscript that addresses the points raised during the review process.

We would appreciate receiving your revised manuscript by Mar 01 2020 11:59PM. To enhance the reproducibility of your results, we recommend that if applicable you deposit your laboratory protocols in protocols.io, where a protocol can be assigned its own identifier (DOI) such that it can be cited independently in the future. For instructions see: http://journals.plos.org/plosone/s/submission-guidelines#loc-laboratory-protocols

We look forward to receiving your revised manuscript.

Kind regards,

Anirudh K. Singh, Ph.D

Academic Editor

PLOS ONE

Journal Requirements:

Reviewers' comments:

Reviewer's Responses to Questions

**Comments to the Author**

1. Is the manuscript technically sound, and do the data support the conclusions?

Reviewer #1: Partly

Reviewer #2: Yes

2. Has the statistical analysis been performed appropriately and rigorously? 

Reviewer #1: N/A

Reviewer #2: N/A

3. Have the authors made all data underlying the findings in their manuscript fully available?

Reviewer #1: No

Reviewer #2: Yes

4. Is the manuscript presented in an intelligible fashion and written in standard English?

Reviewer #1: No

Reviewer #2: No

5. Review Comments to the Author

Reviewer #1: Dear authors,

Please find detailed review comments for the manuscript in attached folder. It need to be rewritten for better understanding of audience with correction in english. The approach is good but scientific finding need to be stated clearly. Figures need to be improved, its hard to read or visualise.

Reviewer #2: In this manuscript titled “Weirdo19ES is a ……………………………Mycobacterium smegmatis mutants” Suarez and co-workers describe the genomic features of a singleton Mycobacteriophage Weirdo19ES by using various bioinformatics tools. They further described the possible receptor for this phage on M. smegmatis surface and demonstrated that cell surface polar GLPs act as receptor. I must admit that authors have done a very thorough analysis of this new mycobacteriophage and their study is well designed and executed. However, there are few issues with the manuscripts;

Major issues:

1. While the M. smegmatis GLP deficient mutants are resistant to Weirdo19ES the experiments performed here do not establish direct interaction of Phage to GLPs on bacterial cells. It will be advisable to do an experiment where Phage is adsorbed on purified GLPs before being used in the plaque assay. In the absence of sufficient quantity of purified GLPs same experiment can be performed using heat killed 1R-Weirdo19ES and wild type MC2155.

2. The manuscript needs careful proofreading for correcting grammatical and typographical mistakes. I have highlighted some of the examples in the minor issues section in the attachment. However, this is not the comprehensive list of language related mistakes. I strongly encourage the authors to have the manuscript proofread for language from their English proficient colleagues.

Minor issues:

Please refer to the attachment.

6. PLOS authors have the option to publish the peer review history of their article (what does this mean?). If published, this will include your full peer review and any attached files.

Reviewer #1: Yes: Vandana Singh

Reviewer #2: No

---

## [Author Response · Author response to Decision Letter 0]

10 Mar 2020

PONE-D-19-31407

Reply to Reviewer 1

Weirdo19ES is a novel singleton mycobacteriophage that selects for glycolipid deficient phage resistant Mycobacterium smegmatis mutants

This is a nice attempt by Cristian Alejandro Suarez and coauthors to characterize singleton mycobacteriophage that selects for glycolipid deficient phage resistant Mycobacterium smegmatis mutants. The authors have compiled large set of bioinformatics data for mycobacteriophage. This characterized mycobacteriophage Weirdo19ES is adding new information and opening research possibilities on the role of mycobacterial GPLs as receptors. On through review of the manuscript, I found several errors which need to be corrected before acceptance for publication in the journal.

Comments: 

The english is very poor, and grammatically incorrect. It’s hard to understand the statements made by authors, as those are too long and complex. Please be consistent in using nomenclature (Use Weirdo19ES or 19WeirdoES, similarly Maximum Likelyhood method or Maximum Likelihood method). Through rewriting with grammar corrections is required for possible acceptance. 

Authors reply: We deeply apologize as the reviewer was right, there were many mistakes that went undetected. We have improved the writing by making shorter sentences where possible. We have corrected the discrepancies on the nomenclature. 

• Figures are with compromised resolutions, hardly readable and difficult to understand the message. Fonts and figures are not appropriate for publication.

Authors reply: Regarding resolutions, all figures have passed the test requested by the journal submission platform (PACE), and after careful review, we found that they have the fonts required by the Journal. However, we do agree that due to the amount of information some figures may be hard to read. Several of the figures presented in this manuscript are similar (due to the common bioinformatics programs used) to others we previously published in this journal. Comparable figures are often presented by other research groups working on mycobacteriophages. We will address the nature of the problems with fonts and resolution with staff members of PlosOne if needed.

In line 226, 70,19% GC content need to be corrected with 70.19%. Similarly, in line 227, 96,1% should be 96.1%. 

Authors’ reply: We apologize for the mistake which has been corrected.

Analysis and homology of the LysA needs to be revalidated. In Fig 2A, both Che9C and DS6A are similar in domain organization, as they belong to same group Org-X. But in Fig. 2B, homology has been studied only with DS6A and other genomes? Why not Che9C is included in this study? If it was included, what is the homology with Weirdo19ES? If it has homology, then why not to say that Weirdo19ES is similar to Org-X group. In Fig. 2B, it looks like SG4 and some other genomes also have homology with Weirdo19ES. Please include these other homology. 

Authors’ reply: We apologyze as we included Che9c instead of Che9d by mistake. That point has been fixed. Fig 2A represents a depiction of some of the most frequent domain combinations found in mycobacteriophage Lys A enzymes. Homology analysis shown in Fig 2B is made on the C-end of those Lys A enzymes that showed highest homology to Weirdo19ES Lys A C-terminus. The reviewer mentions that other genomes have homology to Weirdo19ES however we were only comparing a fragment of LysA and not whole genomes.

In line 269, what is the significance of “Nonstructural genes ORFs 33-40”?

Authors’ reply: This line has been corrected to ensure proper understanding. The sentence refers to genes encoding ORFs that do not bear homology to those playing structural roles in virion structure. The designation is widely used and we feel that we are using the appropriate nomenclature.

In genome organization (line 234), ORF 33-40, 43, 44, 62 and 77 are mentioned to be transcribed leftward. There is no information about gene 45 in this part, but in integration and immunity region (line 283), gene 45 is included as a part of ORF 43-45. What is the orientation of gene 45, it is leftward or rightward?

Author’s reply: As mentioned in the text and in this revised versions genes 43 and 44 are oriented leftwards and gene 45 rightward.

Statement in line 235 and 236 is that gene 43 (int) and gene 44 (rep) are transcribed leftward, while in line 314, 315 and 316, statement tells that int and rep are transcribed rightward. This kind of statements are due to shear negligence and must be taken care prior submission.

Authors reply: Fixed, the correct orientation is shown also in Fig 1 originated through Phamerator.

In line 285, phages BPs, Halo, Annihilator and Avrafan are mentioned having relationship with Weirdo19ES, but in Fig.3A, Annihilator is missing. 

Authors reply: Mycobacteriophages of G1 cluster are numerous and display extensive homology. We have chosen a few of them to illustrate the analysis of integrase and excisionase genes. Once again, the authors would like to mention that a set of phages were included in each analysis to avoid a heavy crowding of sequences and names in the figures. In each case, the set of phages was randomly chosen but representative of a major group displaying homology at the level of genomes or any specific gene under analysis.

In Fig. 3A on the basis of integrase, homology is reported with BPs, Halo, Annihilator, Avrafan, UmaThurman, cluster CVs and Y. In Fig. 4, only BPs, UmaThurman and Bipper are included for intervening region and ssrA tag study. What was the basis to select these members only? 

Authors’ reply: Only one phage from each cluster was picked on the basis of the highest homology for the integrase or the repressor to those of phage Weirdo19ES. In the case of repressor analysis, that of phage UmaThurman was included in spite of being the less homologous one to those of phage Weirdo19ES as it is a Gordonia infecting phage and was included in the previous analysis of integrases.

Similarly, on the basis of repressor, homology/ similarity was with cluster Y, not with G1. Bipper is from cluster Y, but in Fig.4B, BPs also have that conserved alanine. What is the conclusion of this observation?

Authors’ reply: For the analysis of repressor and integrases we picked only one phage of each cluster displaying the highest homology to repressor and integrase of Weirdo19ES. It is not clear to us what is the question raised by the reviewer. 

In Fig. 4B, colour of highlight is not visible.

Authors’ reply: Fixed

Heading in line no 334 has error.

In line 340, ORF 89 is predicted to be HNH using InterProScan. In line 345, ORF 61 is proposed to be HNH, using HMMER. It is not clear that both ORFs (89 and 61) are HNH? Why authors used two different platforms and predicted two different HNH.

Authors´ reply: The use of more than one program to pinpoint specific features, in this case HNHs domains, helps to increase the possibility of detection. Nonetheless, the confirmation of the identity of HNHs will require biological confirmation. 

In comparative genomics study, why group Y (significantly similar in repressor based study) was not included and why K2, Q, D1, G2, G3 and G4 were included as these were not included or mentioned in any other study of the article.

Authors’ reply: Comparative genomics has been carried out with the closest related whole bacteriophage genomes while Rep studies were carried out only based on those temperate phages displaying homology to ours.

In line 411, authors stated that Weirdo19ES, Q and Z are more related to CV & D1. But what about intermediate values of G1, G4, G2, G3?

According to authors G is distant, it is surprising as it has lower values than CV. 

Authors’ reply: Fig 7A displays the GCD of each cluster of phages vs Weirdo19ES while Fig 7B shows GCD contents of all vs all phages (including Weirdo19ES). We looked into GCD content of mycobacteriophages and phages infecting Gordonia in an attempt to find out whether Weirdo19ES would be more related to Gordonia, however, as GCD values for Gordonia phages or mycobacteriophages are not clustered when compared to Weirdo19ES (Fig 7A) it is not possible identify a whether our phage was originally infecting either Gordonia or mycobacteria. 

So, intermediate values of other G subclusters are not relevant for our purpose.

In line 465, it is 102 CFU or 102 CFU?

Authors’ reply: 102. The mistake has been corrected

Legend of Fig.11 says GLP, while Figure denotes GPL. 

Authors´reply: Fixed

Pictures are not clear in Fig.11 A and B. What was the loading control?

Authors’ reply: Fig. 11 is a typical picture of lipid bands after charring, blurring is unavoidable. If by loading control the reviewer refers to mass, comparable volumes of lipid extracts obtained from comparable culture cell pellets were loaded, thus all lanes should be comparable in terms of contents.

Conclusion portion must be summarised. 

Authors’ reply: We have rewritten most of the text trying to be concise but not at the expense of a reasonable presentation and discussion of results. We hope this new manuscript will be acceptable for the Reviewers.

 

 Reply to Reviewer 2

In this manuscript titled “Weirdo19ES is a ……………………………Mycobacterium smegmatis mutants” Suarez and co-workers describe the genomic features of a singleton Mycobacteriophage Weirdo19ES by using various bioinformatics tools. They further described the possible receptor for this phage on M. smegmatis surface and demonstrated that cell surface polar GLPs act as receptor. I must admit that authors have done a very thorough analysis of this new mycobacteriophage and their study is well designed and executed. However, there are few issues with the manuscripts;

Major issues: 

1. While the M. smegmatis GLP deficient mutants are resistant to Weirdo19ES the experiments performed here do not establish direct interaction of Phage to GPLs on bacterial cells. It will be advisable to do an experiment where Phage is adsorbed on purified GLPs before being used in the plaque assay. In the absence of sufficient quantity of purified GLPs same experiment can be performed using heat killed 1R-Weirdo19ES and wild type MC2155. 

Authors’ reply: Although the reviewer raises a reasonable point, the fact that mutants deficient in GPLs both by transposon insertion (and thus bearing genetically identified lesions) and spontaneous resistant mutants, become resistant to the phages under analysis strongly suggest that GPLs are part of the natural receptors for the phages. However, as commented in the Discussion section, an approach comparable to that suggested by the reviewer has been undertaken at early times when there were no tools available for genetic manipulation of mycobacteria. Since this is a microbiology lab we were able to show differences in GPL contents in wild-type and phage resistant mutants, however, GPL extraction and species purification in amounts enough to perform the requested experiment is not within our capabilities. GPLs are the outermost components of the mycobacterial cell envelope and thus the first logical choice to be a receptor. 

The use of heat killed phage resistant cells will not show any interference with phage binding to the wild type strain. Conversely, heat killed wild-type cells may interfere. However, cell envelope components (mainly GPLs and fatty acids) of M. smegmatis are usually shed to the culture medium specially when Tween is used to achieve smooth growth. Tween inhibits phage adsorption, thus we should need to wash cells and with that loose possible phage receptors in the wash step. Thus, although we do appreciate the insightful suggestion, we preferred to use a genetic approach as shown in the manuscript.

2-The manuscript needs careful proofreading for correcting grammatical and typographical mistakes. I have highlighted some of the examples in the minor issues section below. However, this is not the comprehensive list of language related mistakes. I strongly encourage the authors to have the manuscript proofread for language from their English proficient colleagues. 

Authors reply: We deeply apologize, there were a number of grammar mistakes and typos that went undetected. The manuscript has been thoroughly checked by the senior author and an English public translator. An intriguing issue is the fact that figures were processed according to PlosOne specific guidelines using the PACE software and they were approved before uploading the manuscript. If the problem persists we will ask the editorial staff to help us out.

We hope this version will be acceptable and again thank you for your suggestions

Minor issues:

Please remember this is a small list of typographical and grammatical mistakes and the entire manuscript needs to be carefully read and mistakes need to be corrected. 

1. Throughout the manuscript punctuations after the reference are added after space, remove them. 

2. At several places a hyphen (-) is added before a word for no reason, remove them.

3. At some places a comma (,) is used instead of the decimal (.)

4. Line 41: mycobacteriophage Patience from patient

5. Lines 43-44: In the same scenario, analysis of phages infecting Gordonia species showed similarities to mycobacteriophages [4] (not adding anything)

6. Line 47: It called caught our attention the fact that uncommon

7. Line 79: Preparation and of

8. Line 80: briefly late log fresh phase

9. Line 133: Is it true that the images for colony morphology were taken at 0.8X magnification? Or should it be 8X?

10. Line 152: Was 0.5% glycerol in the medium was taken by weight (w/v) or is it a typographical mistake? 

11. Line 168: Provide reference for Jacobs and Hatfull protocol.

12. Line 212: The To this

13. Line 334: genome if of Weirdo19ES

14. Change the title of the last section from Conclusion to Discussion and remove Discussion from Results and Discussion title.

15. Title for Fig. 9 should be “Phage resistant mutants of Mycobacterium smegmatis”

16. Line 468: 5 ml drop seems do big to be used for this experiment. It must be a typographical mistake, correct it.

17. In the legends of Fig. 9 and Fig. 10 the phage is written as 19WeirdoES, match it to nomenclature used in the rest of the manuscript i.e. Weirdo19ES.

18. Include a color key of pham in Fig. 1.

19. Include the schematic of typical LysA with all the domain as a reference.

20. Images attached are blurred, provide good quality images for review. 

Authors’ reply: We believe we have corrected the manuscript. Of note , in Line 41 of the original submission, Patience is the name of the phage so it was mantained as such.

18: pham color assignation is by default

19. As stated in the text there is no typical LysA

---

## [Decision Letter · Decision Letter 1]

3 Apr 2020

Weirdo19ES is a novel singleton mycobacteriophage that selects for glycolipid deficient phage-resistant Mycobacterium smegmatis mutants

PONE-D-19-31407R1

Dear Dr. Morbidoni,

This mail should cheer you up in this time COVID-19 distress. We are pleased to inform you that your manuscript has been judged scientifically suitable for publication and will be formally accepted for publication once it complies with all outstanding technical requirements.

With kind regards,

Anirudh K. Singh, Ph.D

Academic Editor

PLOS ONE

Additional Editor Comments (optional):

Reviewers' comments:

Reviewer's Responses to Questions

**Comments to the Author**

1. If the authors have adequately addressed your comments raised in a previous round of review and you feel that this manuscript is now acceptable for publication, you may indicate that here to bypass the “Comments to the Author” section, enter your conflict of interest statement in the “Confidential to Editor” section, and submit your "Accept" recommendation.

Reviewer #1: All comments have been addressed

Reviewer #2: All comments have been addressed

2. Is the manuscript technically sound, and do the data support the conclusions?

Reviewer #1: Yes

Reviewer #2: Yes

3. Has the statistical analysis been performed appropriately and rigorously? 

Reviewer #1: N/A

Reviewer #2: N/A

4. Have the authors made all data underlying the findings in their manuscript fully available?

Reviewer #1: Yes

Reviewer #2: Yes

5. Is the manuscript presented in an intelligible fashion and written in standard English?

Reviewer #1: Yes

Reviewer #2: Yes

6. Review Comments to the Author

Reviewer #1: Please take care of grammar and do needful revision before final submission. Have a critical look on Fig.3 B about clustering analysis of repressors.

Reviewer #2: The manuscript has improved improved significantly after the revision. The authors have addressed all the reasonable queries.

7. PLOS authors have the option to publish the peer review history of their article (what does this mean?). If published, this will include your full peer review and any attached files.

Reviewer #1: Yes: Vandana Singh

Reviewer #2: No

---

## [Editor Report · Acceptance letter]

6 Apr 2020

PONE-D-19-31407R1 

Weirdo19ES is a novel singleton mycobacteriophage that selects for glycolipid deficient phage-resistant *M. smegmatis* mutants 

Dear Dr. Morbidoni:

I am pleased to inform you that your manuscript has been deemed suitable for publication in PLOS ONE. Congratulations! Your manuscript is now with our production department. 

With kind regards,

on behalf of

Dr. Anirudh K. Singh 

Academic Editor

PLOS ONE